# Predicting the Generalization Ability of a Few-Shot Classifier

**Myriam Bontonou** [1,*], **Louis Béthune** [2], **Vincent Gripon** [1]

[1] Département Electronique, IMT Atlantique, 44300 Nantes, France; vincent.gripon@imt-atlantique.fr
[2] Université Paul Sabatier, 31062 Toulouse, France; louis.bethune@univ-toulouse.fr
[*] Correspondence: myriam.bontonou@imt-atlantique.fr

**Abstract:** In the context of few-shot learning, one cannot measure the generalization ability of a trained classifier using validation sets, due to the small number of labeled samples. In this paper, we are interested in finding alternatives to answer the question: is my classifier generalizing well to new data? We investigate the case of transfer-based few-shot learning solutions, and consider three settings: (i) supervised where we only have access to a few labeled samples, (ii) semi-supervised where we have access to both a few labeled samples and a set of unlabeled samples and (iii) unsupervised where we only have access to unlabeled samples. For each setting, we propose reasonable measures that we empirically demonstrate to be correlated with the generalization ability of the considered classifiers. We also show that these simple measures can predict the generalization ability up to a certain confidence. We conduct our experiments on standard few-shot vision datasets.

**Keywords:** few-shot learning; generalization; supervised learning; semi-supervised learning; transfer learning; unsupervised learning

## 1. Introduction

In recent years, Artificial Intelligence algorithms, especially Deep Neural Networks (DNNs), have achieved outstanding performance in various domains such as vision [1], audio [2], games [3] or natural language processing [4]. They are now applied in a wide range of fields including help in diagnosis in medicine [5], object detection [6], user behavior study [7] or even art restoration [8].

In many cases, the problem at hand is a classification problem. It consists in learning to label data using a set of training examples. In practice, there is a risk of overfitting [9], that is to say that the classifier may focus on certain specificities of the training examples and not be able to generalize well to new ones. That is why its ability to generalize is often evaluated on another set of labeled examples called validation set [10].

Problematically, stressing the generalization ability of a classifier using a validation set requires having access to a large quantity of labeled data. Yet annotating data typically costs money or requires the help of human experts. Even more inconvenient, in some cases the acquisition of data is in itself costly. An extreme case is when one has only access to a few labeled samples, referred to as few-shot [11,12] in the literature. In such a case, trained classifiers are even more likely to suffer from overfitting due to the small diversity of training data. The non-accessibility to a validation set to evaluate generalization becomes thus even more critical.

The objective of this paper is to address the following problem: **can the generalization ability of a few-shot classifier be estimated without using a validation set?**

Throughout this work, we propose to experimentally study several measures. We select the ones that are the most correlated with the performance of few-shot classifiers on a validation set. Then, we evaluate the ability of the selected measures to predict the difficulty of a few-shot task.

In the Experiments reported in Section 5, we only consider the case of transfer-based few-shot classifiers (no meta-learning). These solutions are typically based on the idea of

training generic feature extractors from large generic datasets, which are then used directly as a preprocessing step when facing a new few-shot task. We study multiple few-shot learning classification problems (varying the size of the training set and the number of classes) under 3 settings (supervised, semi-supervised and unsupervised).

Our work comes with the following contributions:

- To the best of our knowledge, we propose the **first benchmark of generalization measures in the context of transfer-based few-shot learning**.
- We conduct experiments to stress the ability of the measures to correctly predict generalization using different settings related to few-shot: (i) supervised, where we only have access to a few labeled samples, (ii) semi-supervised, where we have access to both a few labeled samples and a set of unlabeled samples and (iii) unsupervised, where no label is provided.

The paper is organized as follows. Section 2 is dedicated to a review of the related work. In Section 3, we introduce the formalism and methodologies of few-shot learning. In Section 4, we propose measures to predict the accuracy of a classifier trained with few labeled samples. The measures are assessed in the experiments of Section 5 (Code at https://github.com/mbonto/fewshot_generalization). Section 6 is a conclusions.

## 2. Related Work

In this paper, we address the difficulty of predicting the generalization ability of a few-shot classifier. In this section, we detail some of the works related to this issue.

### 2.1. Few-Shot Learning

As training a DNN on few data samples from scratch typically leads to overfitting, other learning strategies have been developed. All these strategies share the idea of building a general-purpose representation of the data. In the literature, many strategies are based on transfer learning, where a DNN, called *backbone*, is pretrained on a huge annotated dataset. The backbone is used as a feature extractor for the few-shot task. The huge dataset is composed of what are called *base classes* whereas the classes considered in the few-shot task are called *novel classes*.

#### 2.1.1. With Meta-Learning

A first group of strategies uses meta-learning. It consists in using entire tasks as training examples. Some *optimization-based* works propose to learn a good initialization of the weights of the DNN over several training tasks, so that a new task can be learned with only a few gradient steps [13,14]. In *metric-based* works [15–19], the idea is to learn to embed the data samples in a space where classes are easily separable. Thus, once a new task occurs, the features of the novel samples are projected into this embedding (without any learning) and a simple classifier is trained to recognize the novel classes from these features. As the number of parameters to learn is reduced, the risk of overfitting is lower. There are many variants in the literature. For instance, in [15], the authors assume that there is an embedding space where each class is represented by one point. Thus, a DNN is trained over several training tasks to work with a distance-based classifier, in which each class is represented by the average of its projected data samples. When a new task comes, the representations of the samples are extracted from the DNN, and the labels of the new samples are attributed according to the class of the closest representative.

#### 2.1.2. Without Meta-Learning

In a recent line of work, some methods do not focus on learning a relevant embedding from training tasks but on learning a relevant embedding from a single classification task involving all training classes at once [11,12,20,21]. First, a DNN is trained to minimize a classification loss on base classes. A regularization term such as self-supervision [11,21] or Manifold Mixup [11] is sometimes added to the loss to learn more robust features. Then, the features of the samples of the few-shot task are extracted from the DNN (often using the

features of its penultimate layer, just before the classifier). Finally, a simple classifier, such as a Nearest Class Mean [12] or a Cosine Classifier [11], is trained on the extracted features to distinguish between classes. In [12], the authors show that simple transformations, such as normalizing extracted features with their $L_2$-norm, help the classifier generalizing better. Using self-supervision and Manifold Mixup, the article [11] achieves state-of-the-art performance on benchmark datasets. That is a first reason explaining why, in this article, we allow to restrict our study to few-shot learning solutions based on pretrained feature extractors. The second reason is about computational constraints: unlike meta-learning methods, the same DNN can be reused without additional training in all experiments.

### 2.2. Better Backbone Training

In transfer-based few-shot learning, the challenge is to learn representations on training classes which are suitable for novel classes. Indeed, the generalization ability of a classifier is linked to the distribution of the representatives of the data samples in the feature space. However, it is not easy to estimate whether the learned embedding space suits novel classes well.

#### 2.2.1. Learning Diverse Visual Features

The generalization ability of the classifiers depends on the relevance of the extracted features for a new task. Inspired by works in deep metric learning, the authors of [22] propose to learn representations capturing general aspects of data. They optimize a DNN to perform a range of tasks enhancing class-discriminative, class-shared, intra-class and sample-specific features. Although they do not apply their method to few-shot tasks, it could help improving the generalization. Similarly, self-supervised learning and Manifold Mixup used in [11] improve the performance on few-shot tasks.

#### 2.2.2. Using Additional Unlabeled Data Samples

Another way to learn richer representations is to use additional unlabeled samples. Any unlabeled sample can be used to to better separate the novel classes by inferring more adapted representations of the data. In the literature, two settings are studied. In the setting we consider in this article, the unlabeled samples are the samples on which the accuracy of the classifier will be evaluated. In the other setting, the unlabeled samples are just seen as additional samples and they are not used to test the classifier. In [23], both settings are considered. The researchers look for a linear projection which maximizes the probability of being in the correct class. More precisely, first an unsupervised low-dimensional projection (PCA or ICA) is applied on the features to reduce their noise. Then, the samples are clustered using either a Bayesian $K$-Means or a Mean-Shift approach followed by a NCM classifier. In [24], the features of the data samples are diffused though a similarity graph computed from the few-shot samples and from the unlabeled samples before being used in a classifier. As these works use additional information, the generalization performance is increased.

#### 2.2.3. Learning Good Representations

Learning efficient representations has always been a concern for deep learning [25]. Invariant Risk Minimization [26] and $v$-Information [27] have been proposed as theoretical frameworks to detail the properties a good representation should exhibit when connected to a (mostly linear) classifier. Other works focus on maximizing the mutual information (following InfoMax principle) such as Deep Infomax [28]. Losses (like in [22] or in [29]) are designed to enforce some geometry in latent space based on similarity measures. Robust few-shot learning for user-provided data [30] is proposed to handle outliers within training samples.

### 2.3. Evaluating the Generalization Ability

The generalization ability of a few-shot classifier can be improved by designing more relevant representations of data. However, this ability is ill-evaluated in few-shot learning. Indeed, in standard deep learning, the generalization ability is usually estimated on a validation set. Here in few-shot learning, we do not have enough samples to afford to split the training data into a training set and a validation set. Thus, the question of interest in this study, which has not been handled so far in the few-shot literature, is not how to improve the generalization performance but really how to evaluate it. Note that some authors have also been interested in the possibility of evaluating the generalization without relying on a validation set. For instance, in [31,32], the authors try to understand the generalization of DNNs from the training set and the DNNs parameters. Contrary to these studies, we do not consider the backbones parameters or the data on which they have been trained. We only look at the features distribution of the few-shot task samples obtained from the backbones.

### 3. Background

This section is intended as a reference for the rest of the article. The formalism behind few-shot classification is provided. The three considered settings—supervised, semi-supervised and unsupervised, are detailed. For each setting, the studied classifiers are also introduced.

### 3.1. Few-Shot Classification: A Transfer-Based Approach

In this work, we only study the few-shot solutions in which a DNN, pretrained on a large training dataset, is used to extract general-purpose features from the data of the considered few shot task. This network is usually called the *backbone*. The classes used to train the backbone are called *base* classes.

Once the backbone has been trained, we are facing a few-shot task where the objective is to learn to discriminate between *novel classes*, provided only a few samples of those. In the following, we denote $C_b$ the number of base classes and $C_n$ that of novel classes. Note that typically $C_n \ll C_b$.

The backbone can be formalized as a function $g$, such that $g = c_{W_b} \circ f_\theta$. See Figure 1 for an illustration. In our case, $c_{W_b}$ is a $C_b$-way classifier whose parameters are $W_b$ and $f_\theta$ is a convolutional Neural Network. Note that typically $f_\theta$ outputs the penultimate representation within the backbone when processing an input element. Denoting $\mathbf{x}$ a data sample from the few-shot task, its features $\mathbf{f} \in \mathbb{R}^d$ are extracted as follows: $\mathbf{f} = f_\theta(\mathbf{x})$. The features $f_\theta(\mathbf{x})$ are part of the *feature space* $\mathcal{F}$. To solve the few-shot task, a $C_n$-way classifier is trained on top of the extracted features. The predicted label associated with a data sample is denoted $\tilde{y}$ while its true label is denoted $y$.

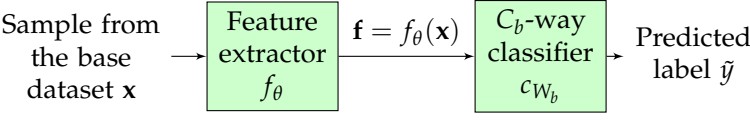

**(a)** Pretraining of the backbone $g = c_{W_b} \circ f_\theta$ on the base classes.

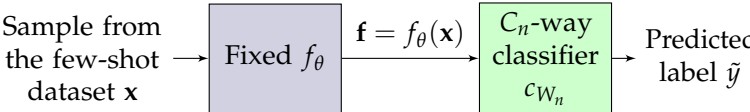

**(b)** Training of the classifier $c_{W_n}$ on the novel classes.

**Figure 1.** Illustration of the transfer-based few-shot learning principle.

### 3.2. Studied Settings

In our work, we consider three different settings: a supervised setting, an unsupervised setting and a semi-supervised setting.

In the supervised setting, we consider *N*-way *K*-shot tasks. The number *N* refers to the number of classes to discriminate from, and *K* to the number of labeled samples we are given for each class. Denote by $\mathbf{x}_{ij}$ the *i*-th labeled element of class *j*, then the training set is written as:

$$S^{\text{supervised}} = \left\{ (\mathbf{x}_{ij}, y_j) | 1 \le i \le K, 1 \le j \le N \right\}. \tag{1}$$

In the unsupervised setting, we consider *N*-way *Q*-query tasks. We are given *Q* unlabeled samples for each of the *N* considered classes. Considering that $\perp$ refers to the fact the labels are unknown, the training set is written as:

$$S^{\text{unsupervised}} = \left\{ (\mathbf{x}_{ij}, \perp) | 1 \le i \le Q, 1 \le j \le N \right\}. \tag{2}$$

Finally, in the semi-supervised setting, we consider *N*-way *K*-shot *Q*-query tasks. Some training samples are labeled, others are not. The training set is:

$$S^{\text{semi}} = S^{\text{supervised}} \cup S^{\text{unsupervised}}. \tag{3}$$

For evaluation purposes, in all settings, there are as many unlabeled samples per class. This methodology is used in several research papers (e.g., in [23,24]).

### 3.3. Studied Classifiers

We now introduce the classifiers we consider in the scope of this paper.

#### 3.3.1. Supervised Setting

A classifier that performs well in many cases in the supervised setting is the logistic regression (LR). Given the number of features *d*, the weight matrix $\mathbf{W} \in \mathbb{R}^{d \times N}$ and the matrix containing the features of all data samples $\mathbf{F} \in \mathbb{R}^{(NK) \times d}$, the output of the LR, denoted $\mathbf{P} \in \mathbb{R}^{(NK) \times N}$, is:

$$\mathbf{P} = \text{softmax}(\mathbf{FW}), \tag{4}$$

where $\mathbf{p}[i, c]$ is the probability that the sample *i* belongs to class *c*. Weights are learned using backpropagation to minimize a cross-entropy loss.

#### 3.3.2. Semi-Supervised Setting

In the semi-supervised setting, we consider a recent classifier reaching state-of-the-art performance on competitive benchmarks [24]. The features extracted from a backbone are diffused through a cosine similarity graph $\mathcal{G}$ before being processed by a usual LR. In the following, the classifier is described more formally.

Let us point out that, as in all backbones we use the features are extracted after a ReLU function, all vectors in $\mathcal{F}$ contain non-negative values. Consequently, the output of the cosine similarity function ranges from 0 (orthogonal vectors) to 1 (aligned vectors). Given $\mathbf{f}_i, \mathbf{f}_j \in \mathcal{F}$, the cosine similarity function is defined as:

$$\cos(\mathbf{f}_i, \mathbf{f}_j) = \frac{\mathbf{f}_i^\mathsf{T} \mathbf{f}_j}{\|\mathbf{f}_i\|_2 \|\mathbf{f}_j\|_2}. \tag{5}$$

The cosine similarity graphs $\mathcal{G} = \langle \mathcal{V}, \mathcal{E}, \mathbf{W} \rangle$ consist in a set of vertices $\mathcal{V}$ connected by a set of edges $\mathcal{E}$. The weights of the edges are stored in the adjacency matrix $\mathbf{W}$. Given two vertices *i*, *j* and their feature vectors $\mathbf{f}_i, \mathbf{f}_j$, the adjacency matrix $\mathbf{W}$ is defined as:

$$\mathbf{W}[i, j] = \begin{cases} \cos(\mathbf{f}_i, \mathbf{f}_j) & \text{if } \{i, j\} \in \mathcal{E} \\ 0 & \text{otherwise} \end{cases}. \tag{6}$$

after removing self-loops, only the $k$-th largest values per row are kept. Then, given the diagonal degree matrix $\mathbf{D}$, the resulting matrix is normalized as follows:

$$\mathbf{E} = \mathbf{D}^{-\frac{1}{2}}\mathbf{W}\mathbf{D}^{-\frac{1}{2}}, \text{ where } \mathbf{D}[i,i] = \sum_j \mathbf{W}[i,j]\,. \tag{7}$$

Given $\mathbf{F} \in \mathbb{R}^{(NK+NQ)\times d}$ the matrix containing the features of all data samples, $\mathbf{I}$ the identity matrix, and two constant $\alpha$, $\kappa$, the new features are obtained by propagating the extracted features as follows:

$$\mathbf{F}_{\text{diffused}} = (\alpha\mathbf{I} + \mathbf{E})^{\kappa}\mathbf{F}\,. \tag{8}$$

More details can be found in the original paper [24].

#### 3.3.3. Unsupervised Setting

The unsupervised setting is less studied in the few-shot literature. We hypothesize that, when features are well adapted to a $N$-way task, each class is associated with a cluster in the feature space. In that case, a standard clustering method consists in using a $N$-means algorithm. In order to compare results with the semi-supervised setting in a fair manner, we also propagate the features extracted from backbones through a cosine similarity graph as detailed in the semi-supervised setting.

## 4. Predictive Measures

Now that we have introduced the classifiers considered in the three settings, the next step is to propose reasonable measures to evaluate their generalization abilities. An overview of the proposed measures is given in Table 1. More details are given in the next paragraphs.

**Table 1.** Table summarizing the solutions considered to predict the generalization ability of a classifier trained on few examples. The solutions are measures designed to quantify how well a trained model generalizes to unseen data.

| | | SETTINGS | | |
|---|---|---|---|---|
| | | ***Supervised N-Way K-Shot*** | ***Semi-Supervised N-Way K-Shot Q-Query **** | ***Unsupervised N-Way Q-Query **** |
| SOLUTIONS | ***Using available labels and features of data samples*** | | | |
| | *Training loss of the logistic regression* | ✓ | ✓ | × |
| | *Similarities between labeled samples* | ✓ | ✓ | × |
| | *Confidence in the output of the logistic regression* | × | ✓ | × |
| | ***Using only data relationships*** | | | |
| | *Eigenvalues of a graph Laplacian* | ✓ | ✓ | ✓ |
| | *Davies-Bouldin score after a N-means algorithm* | ✓ | ✓ | ✓ |

* Query samples are accessible during training without their labels.

### *4.1. Supervised Setting*

In the supervised setting, the classifier we consider is a logistic regression (LR). We propose two measures to estimate how well the trained LR generalizes to unseen data. The first one is the LR training loss obtained at the end of the training process. The second one is based on the similarities between labeled data samples, and is therefore agnostic of the choice of the LR as classifier.

### 4.1.1. LR Training Loss

The LR is trained to minimize the cross-entropy loss, see Definition 1. During training, this loss is supposed to converge to zero. Assuming that harder the task is, slower the convergence is, the value of the loss at the end of the training should give some insights about the difficulty of the task.

**Definition 1** (LR loss). *Given $y_{ic}$ the number (0 or 1) indicating if the label of the data sample i is c and $p_{ic}$ the output of the LR indicating the probability of i being labeled c, the loss is defined as:*

$$LR\ loss = \frac{-1}{NK} \sum_{i=1}^{NK} \sum_{c=1}^{N} y_{ic} \log p_{ic}. \tag{9}$$

### 4.1.2. Similarity

Recent Works [12,24] have shown that state-of-the-art performance can be achieved by comparing distances to a centroid defined for each class. The difficulty of the clustering can be measured by comparing the intra-class similarity to the inter-class similarity. In our case, we have chosen the cosine similarity measuring the cosine of the angle between two vectors (see Equation (5)) because it does not require to define arbitrary parameters (contrary to a RBF kernel). As the feature vectors are extracted from the backbones after a ReLU function, the cosine similarity is naturally between 0 (orthogonal vectors) and 1 (perfectly aligned). This choice is also justified by the fact that in few-shot learning, it is usual to divide the feature vectors by their norms to improve the performances [11,12]. After that preprocessing step, the norms of the feature vectors no longer carry relevant information. The notions of intra-class and inter-classes similarities are defined in Definition 2 and 3. If a class $c$ only contains one shot, we set intra$(c) = 1$. The proposed measure is detailed in Definition 4.

**Definition 2** (Intra-class similarity). *The cosine similarity within a class c is:*

$$intra(c) = \frac{1}{K(K-1)} \sum_{\substack{i \\ y_i=c}} \sum_{\substack{j \neq i \\ y_j=c}} cos(\mathbf{f}_i, \mathbf{f}_j). \tag{10}$$

**Definition 3** (Inter-classes similarity). *The cosine similarity through classes c and $\tilde{c}$ is:*

$$inter(c, \tilde{c}) = \frac{1}{K^2} \sum_{\substack{i \\ y_i=c}} \sum_{\substack{j \\ y_j=\tilde{c}}} cos(\mathbf{f}_i, \mathbf{f}_j). \tag{11}$$

**Definition 4** (Similarity). *The proposed similarity measure is:*

$$similarity = \frac{1}{N} \sum_{c=1}^{N} \left( intra(c) - \max_{c \neq \tilde{c}}(inter(c, \tilde{c})) \right). \tag{12}$$

### 4.2. Unsupervised Setting

In the unsupervised setting, the goal is to estimate the quality of the clustering. To this end, we consider two measures. The first one is a measure of relative similarity between clusters. The other one is an indirect measure of the connectivity of components in a graph whose vertices are training samples and edges represent the similarity between those samples.

### 4.2.1. Davies-Bouldin Score after a N-means Algorithm

Assuming the data samples within the classes to be similar enough, we expect each learned cluster to represent a class. A measure of relative similarity within clusters and

between clusters, such as the classical Davies-Bouldin (DB) score [33], gives an insight about the difficulty of the clustering. Consequently, it may measure how easy it is to generalize to new samples. In Definition 5, we detail the Davies-Bouldin (DB) score. Lower is the score, better is the clustering. It varies between 0 and $+\infty$.

**Definition 5** (DB score). *Denote the centroid of a cluster C $\boldsymbol{\mu}_c$, such that $\boldsymbol{\mu}_c = \frac{1}{|C|} \sum_{i \in C} \mathbf{f}_i$. The average distance between the samples in C and the centroid of their cluster $\boldsymbol{\mu}_c$ is:*

$$\delta_c = \frac{1}{|C|} \sum_{i \in C} \|\mathbf{f}_i - \boldsymbol{\mu}_c\|_2. \tag{13}$$

*Then, the DB score is given as:*

$$DB\ score = \frac{1}{N} \sum_{c=1}^{N} \max_{\tilde{c} \neq c} \left( \frac{\delta_c + \delta_{\tilde{c}}}{\|\boldsymbol{\mu}_c - \boldsymbol{\mu}_{\tilde{c}}\|_2} \right). \tag{14}$$

4.2.2. Laplacian Eigenvalues

Consider a graph where each vertex represents a data sample and where the edges are weighted proportionally to the similarity between the samples. Now, consider that only the biggest edges are kept. In the perfect case where samples from distinct classes are very dissimilar, it is expected that this graph yields at least as many connected components as the number of classes in the considered problem. A measure of the fact that a graph contains at least $N$ connected components is given by the amplitude of the $N$-th lower eigenvalue (egv $N$) of its Laplacian [34]. See Definition 6.

**Definition 6** (Egv N). *We consider the graph $\mathcal{G} = \langle \mathcal{V}, \mathcal{E}, \mathbf{W} \rangle$ where $\mathcal{V}$ is the set of data samples. The adjacency matrix $\mathbf{W}$ is obtained by first considering the cosine similarity between these samples, removing self-loops, and keeping only the k-th largest values on each line/column. The Laplacian of the graph is given by $\mathbf{L} = \mathbf{D} - \mathbf{W}$, where $\mathbf{D}$ is the degree matrix of the graph: $\mathbf{D}$ is a diagonal matrix where $\mathbf{D}_{ii} = \sum_{j=1}^{NQ} \mathbf{W}_{ij}$. The measure we consider is the amplitude of the N-th lower eigenvalue of $\mathbf{L}$.*

*4.3. Semi-Supervised Setting*

In the semi-supervised setting, the classifier we consider is the adapted LR detailed in Section 3. We propose a measure based on the confidence of the LR decision on the unlabeled samples.

The confidence can be obtained by looking at the distance between the provided output and a one-hot-bit encoded version of this output. In more details, for each unlabeled sample, the classifier outputs the probability it belongs to a particular class. As we do not know the label of the sample, we cannot look at the probability the classifier gives to its real class. However, we propose to report the maximal probability the classifier attributes to the classes (see Definition 7). If the maximal probability is far from one, we can interpret it as the classifier is unsure of its output. So, lower the maximal probability is, harder the task should be for the considered sample.

**Definition 7** (LR confidence). *Let $p_{ic}$ denote the probability that the data sample i is labeled c.*

$$LR\ confidence = \frac{-1}{NQ} \sum_{i=1}^{NQ} \log \max_c (p_{ic}). \tag{15}$$

In the next section, we empirically evaluate the relevance of the proposed measures to estimate the performance of the classifiers on new samples.

## 5. Experiments

In this section, we evaluate the ability of the previously proposed measures to predict the generalization of a transfer-based trained few-shot classifier. First, we describe the datasets, backbones and evaluation metrics used throughout the experiments. Then, for each setting, we report the correlation between the proposed measures and the accuracy of classifiers measured on a validation set. In Section 5.7, we look at the more difficult problem of using the proposed measures to predict the performance of a few-shot classifier. In Section 5.8, we propose an experiment where we actively label samples that are predicted to be the hardest to predict, resulting in an overall increased accuracy. Finally, in Section 5.9, we explore less standard settings where the number of samples to predict for each class is unbalanced.

### 5.1. Datasets

We consider two datasets. The first one is mini-ImageNet [16]. It has been generated from the bigger ImageNet database [35]. It is split into 64, 16 and 20 classes, in which 600 images are available. The first split is used to train the backbone, the second to validate its generalization ability and the third one to generate the few-shot tasks. The second dataset is tiered-ImageNet [36]. The splits contain 351, 97 and 160 classes, with roughly about 1000 samples each. It is also extracted from ImageNet. The interest of tiered-ImageNet is that the semantic of classes has been studied with WordNet [37] to ensure that the considered splits contain semantically different classes. In both datasets, as in numerous studies [12], the images are resized to $84 \times 84$ pixels.

When generating a few-shot task, $N$ classes are uniformly drawn at random in the last introduced split. The $K$ and $Q$ samples to generate from each class are uniformly drawn without replacement. To assess the generalization performance in the supervised setting, we measure the performance on 50 samples uniformly drawn from the remaining items for each considered class (that is to say items that were not drawn to be part of the $K$-shot). In the unsupervised and semi-supervised settings, the performance is measured on the $Q$ unlabeled samples per class.

### 5.2. Backbones

We consider two backbones. The first one is a Wide Residual Network [38] (**wideresnet**) of 28 layers and width factor 10 described in [11]. It has been trained on mini-ImageNet with a classification loss (classification error), an auxiliary loss (self-supervised loss) and fine-tuned using manifold mix-up [39]. Its results are among the best reported in the literature. The second backbone is a DenseNet [40] (**densenet**) trained on tiered-ImageNet from [12]. As advised in the original papers, all feature vectors are divided by their $L_2$-norm: given $\mathbf{f} \in \mathcal{F}$, $\mathbf{f} \leftarrow \frac{\mathbf{f}}{\|\mathbf{f}\|_2}$ .

### 5.3. Evaluation Metrics

In the supervised and semi-supervised setting, the performance is evaluated with the accuracy on the query samples. In the unlabeled setting, the quality of the clustering is evaluated with an Adjusted Rand Index (ARI). This index ranges from 0 to 1, 1 meaning that the data samples are exactly clustered according to their labels, and 0 that the clustering is at chance level. As we have observed that the relations between the measures and the performance on the test sets are rather linear (see Figures 3 and 5), we report in Sections 5.4–5.6 the absolute values of the Pearson correlation coefficients between the measures and the performance in the three settings.

### 5.4. Correlations in the Supervised Setting

Considering $N$-way $K$-shot tasks, we look at the linear correlation between the measures and the accuracy of the LR on new samples. Precisely, on 50 data samples per class not used during training. In the case of the unsupervised measures (Egv $N$, DB score), we consider that all training samples are unlabeled. In Figure 2, we perform experiments

on mini-ImageNet ((a) and (b)) and on tiered-ImageNet ((c) and (d)). In (a) and (c), we consider 5-way tasks and depict the evolution of the correlation as a function of the number of shots per class. In (b) and (d), we consider 5-shot tasks and make the number of classes varying. As in 1-shot tasks, the DB score is always 0, we do not report a correlation measure. Note that in Appendix B, the accuracies of the LR are reported.

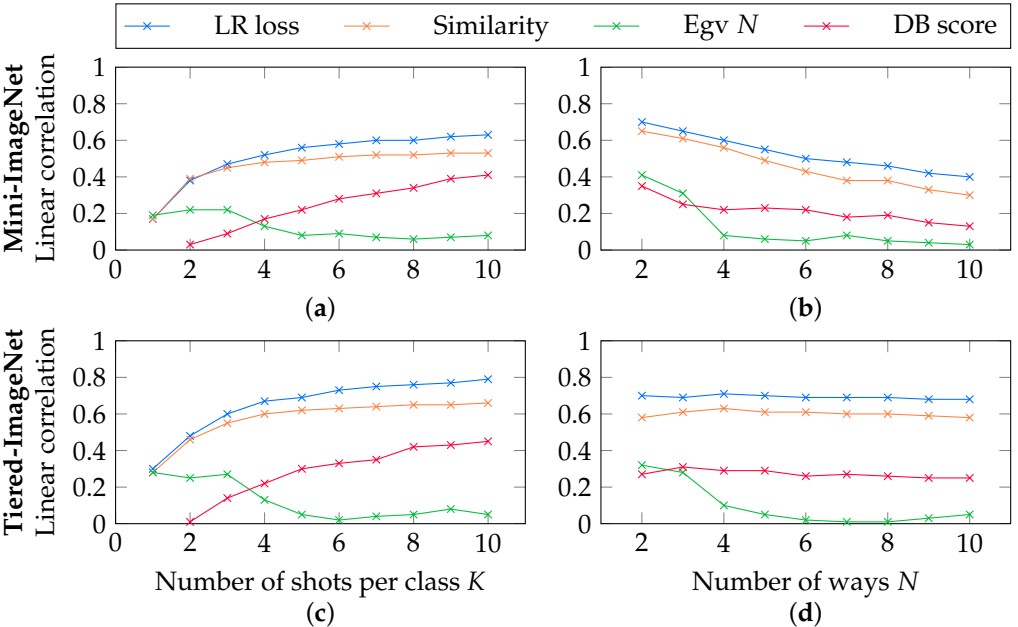

**Figure 2.** Supervised setting. Study of the linear correlations between the measures and the accuracy of a LR computed on a test set. In (**a,b**), the data come from mini-ImageNet. Their features are extracted with **wideresnet**. In (**c,d**), the data come from tiered-ImageNet. Their features are extracted with **densenet**. See Section 5 for details. By default, 5-way 5-shot tasks are generated. In (**a,c**), the number of shots varies. In (**b,d**), the number of classes varies. Each point is obtained over 10,000 random tasks.

In all experiments, we observe that the measures adapted to the supervised setting (i.e., LR loss and Similarity) perform better than the measures designed for an unsupervised setting. That is not surprising because the supervised measures exploits the additional information given by the labels. The best correlation is always obtained with the LR loss. The linear correlation seems to be lower when the tasks are harder (more classes, less shots). The only exception is in the case of the egv $N$ where the correlation decreases with the number of shots. Recall that the egv $N$ measures how well the graph can be split into $N$ communities: zero when there is at least $N$ connected components, and higher when there is no sparse cut into $N$ components. One possible explanation might be that with a higher number of shots, the data tends to be split into more than $N$ communities, which does not damage the performance. Thus, the egv $N$ is always null, which is not informative.

To further investigate what happens, we generate in Figure 3 two plots using data samples from mini-ImageNet. Each point represents a task, with the LR loss on the $x$-axis and the accuracy on the $y$-axis. In (a), 5-way 5-shot tasks are considered. In (b), 5-way 1-shot tasks. In 5-way 5-shot, the relation between both variables is rather linear. Without surprise, in 5-way 1-shot, the LR loss is less representative of the accuracy. With 1 sample per class, it is very hard to detect the hardest tasks.

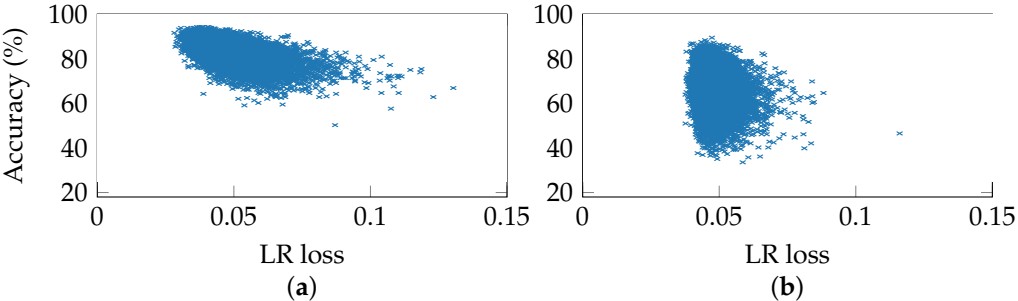

**Figure 3.** Supervised setting. Each point represents a task. We plot the accuracy of a LR in function of the loss of the LR on the training samples. In (**a**), we consider 10,000 random 5-way 5-shot tasks. In (**b**), we consider 10,000 random 5-way 1-shot tasks. The data samples come from mini-ImageNet. Their features are extracted with **wideresnet**.

*5.5. Correlations in the Unsupervised Setting*

Considering $N$-way $Q$-query tasks, we report the linear correlations between the unsupervised measures and the adjusted rand index (ARI) of the $N$-means algorithm. The ARI is computed on the $NQ$ unlabeled samples on which the $N$-means algorithm has been trained. In Figure 4, we perform experiments on mini-ImageNet ((a) and (b)) and on tiered-ImageNet ((c) and (d)). In (a) and (c), we consider 5-way tasks. The number of queries varies. In (b) and (d), we consider 35-query tasks. The number of classes $N$ varies. In Appendix B, the average ARI of the $N$-means algorithm are reported.

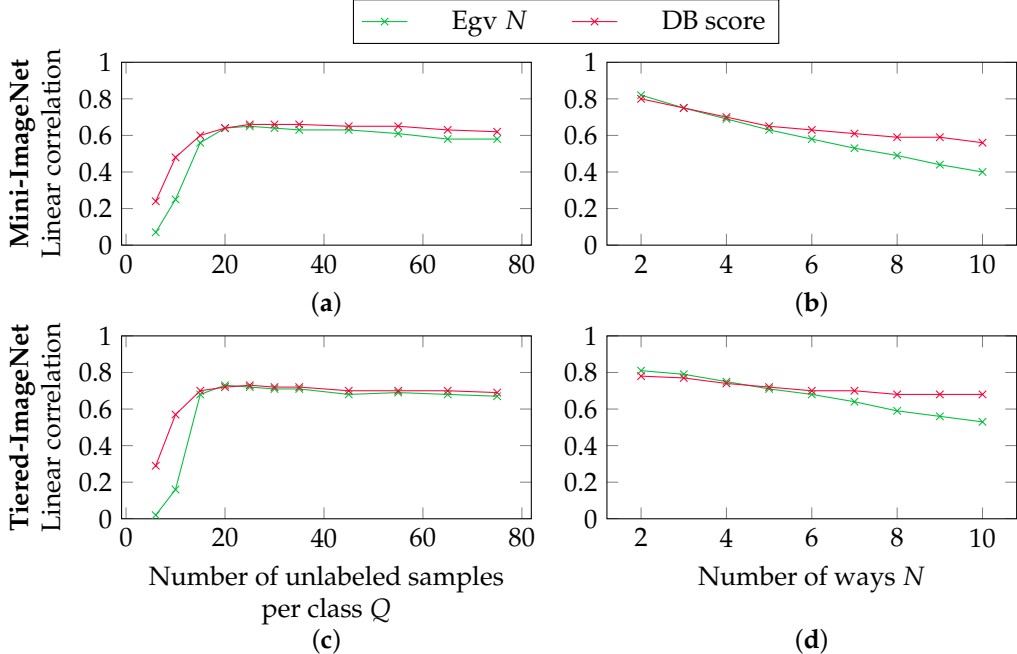

**Figure 4.** Unsupervised setting. Study of linear correlations between the measures and the ARI of a $N$-means algorithm. The ARI is computed on the $NQ$ unlabeled samples available during training. In (**a**,**b**), the data come from mini-ImageNet. Their features are extracted with **wideresnet**. In (**c**,**d**), the data come from tiered-ImageNet. Their features are extracted with **densenet**. All features are diffused through a similarity graph. See Section 3 for details. By default, 5-way 35-query tasks are generated. In (**a**,**c**), the number of queries varies. In (**b**,**d**), the number of classes varies. Each point is obtained over 10,000 random tasks.

In all experiments, the DB-score is the best. Using 20 samples per class enables to increase the correlation up to 0.64 on mini-ImageNet and up to 0.72 on tiered-ImageNet.

As the complexity of the tasks increases (more classes, less queries), our measures are less representative of the problems.

In this experiment, we do not have access to the labels of the samples during the training but, we make sure that each class contains as many samples. In Section 5.9, we explore the impact of an unbalanced distribution. We also propose an additional experiment to see if the $N$-th eigenvalue is really the one giving the best correlation among all other eigenvalues. In Appendix C, we also report an experiment showing the influence of the number of nearest neighbors kept in the graph used to compute the eigenvalue.

### 5.6. Correlations in the Semi-Supervised Setting

We consider $N$-way $K$-shot $Q$-query tasks. The query samples are available without their labels during training. We study the linear correlation between the LR confidence and the accuracy of the LR on the query samples. In practice, on $NQ$ samples. We also look at the correlations obtained with the measures defined on supervised and unsupervised inputs. In the unsupervised case, we consider all training samples as unlabeled samples. In Figure 5, we perform experiments on mini-ImageNet ((a–d)) and on tiered-ImageNet ((e–h)). In (a) and (e), we consider 5-way 5-shot tasks. The number of queries varies. In (b) and (f), we consider 5-way 30-query tasks. The number of shots varies. In (c) and (g), we consider 5-shot 30-query tasks. The number of classes varies. In (d) and (h), there are two scatter plots. Each point represents a task, with the LR confidence on the x-axis and the accuracy on the y-axis. In both cases, 5-way 5-shot 30-query tasks are considered. In Appendix B, the average accuracies of the LR for each data point are reported.

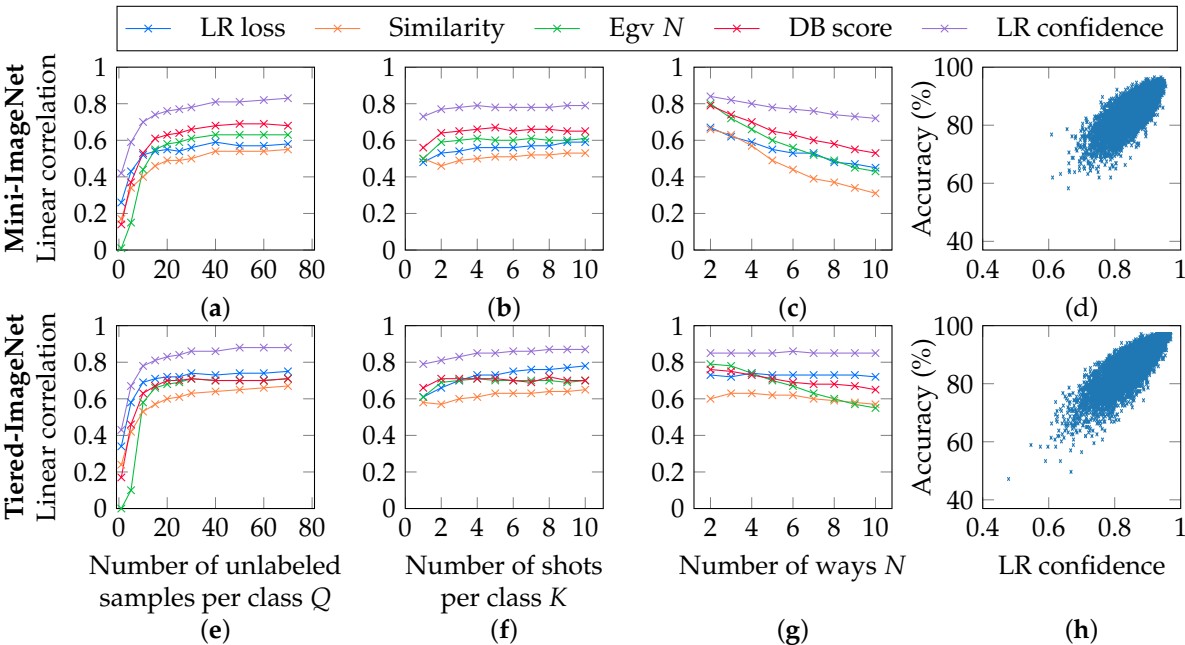

**Figure 5.** Semi-supervised setting. Study of linear correlations between the measures and the accuracy of a LR on the $NQ$ unlabeled samples available during training. In (**a–d**), the data samples come from mini-ImageNet. Their features are extracted with **wideresnet** and diffused through a similarity graph. In (**e–h**), the samples come from tiered-ImageNet. Their features are extracted with **densenet** and diffused though a similarity graph. See Section 3 for details. By default, 5-way 5-shot 30-query tasks are generated. In (**a,e**), the number of queries varies. In (**b,f**), the number of shots varies. In (**c,g**), the number of classes varies. Each point is obtained over 10,000 random tasks. In (**d,h**), each point represents a task. We plot the accuracy of the LR in function of the LR confidence. In (**d**), 5-way 5-shot 30-query tasks are generated from mini-ImageNet. In (**h**), 5-way 5-shot 30-query tasks are generated from tiered-ImageNet.

We observe that the LR confidence, a measure adapted to the semi-supervised setting, outperforms the supervised and the unsupervised measures. The more queries there are, the better the correlation is, with a threshold around 15 additional unlabeled samples per

class. The correlation depends less on the number of shots and classes. The supervised measures seem to depend on the number of unlabeled samples per class $Q$. This is, in part, an artifact due to the fact that the accuracies are computed on the number of queries $NQ$. If $Q$ is small, the range of possible accuracies is reduced, so the computed correlation is affected. We also observe that the correlation between the supervised measures and the accuracy are higher than in the supervised setting for the small $K$. This is due to the fact that the features are previously diffused on a similarity graph.

The LR confidence uses more information than the unsupervised measures (labels) and the supervised measures (more data). As a small number of shots does not reduce the linear correlation, we assume that this is due to the diffusion of the features before the training of the LR.

As in the unsupervised setting, we use as many unlabeled samples per class during training. The impact of an unbalanced distribution is explored in Section 5.9.

### 5.7. Predicting Task Accuracy

In the previous experiments, we focused on a statistical measure of correlation between the proposed measures and the generalization abilities of the considered classifiers. In practice, the question of interest is rather to be able to predict the generalization of a given classifier. To this end, we consider that there are two types of few-shot tasks: hard and easy. The tasks with an accuracy below 80% are considered to be hard and the ones above 80% are easy, 80% being an arbitrary choice. In each setting, we wonder whether the predictive measures enable to distinguish between hard and easy tasks.

In practice, we divide the mini-ImageNet split into two sets. Both containing 10 classes. On both sets, 10,000 5-way 5-shot (30-query) tasks are randomly generated. In the unsupervised case, we consider all training samples as unlabeled samples. In Figure 6, we plot the ROC curve using the first set of tasks. Here, 1—specificity refers to the proportion of tasks predicted as being hard among the easy tasks. The sensibility refers to the proportion of tasks predicted as being hard among the hard tasks. After choosing the threshold value, we report a confusion matrix on the second set. Each row of the confusion matrix is normalized, so that its first (resp. second) row indicates the percentage of tasks predicted hard or easy among the hard (resp. easy) tasks.

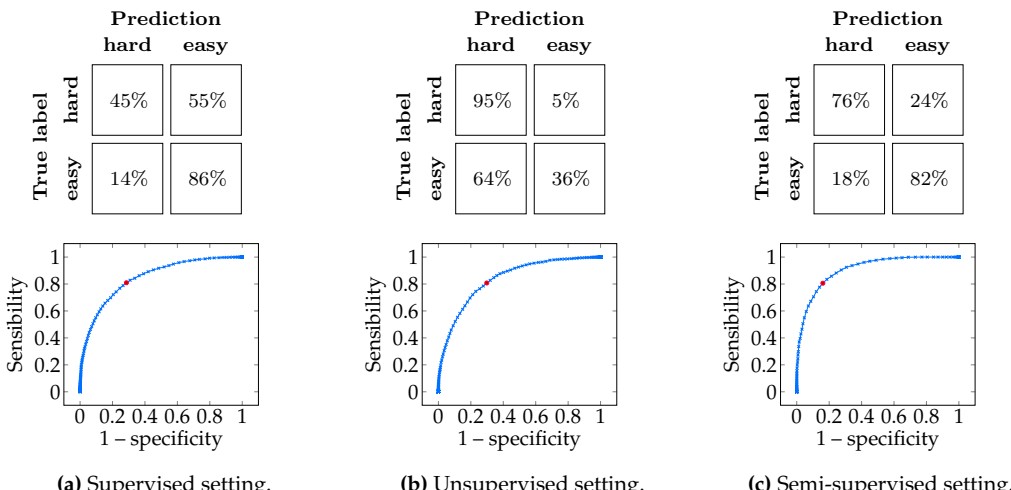

**(a)** Supervised setting.      **(b)** Unsupervised setting.      **(c)** Semi-supervised setting.

**Figure 6.** Task prediction. The ROC curves are computed over 10 classes of mini-ImageNet. The tables are computed on 10 other classes, applying the threshold value denoted by a red point on the curves. Features are extracted with **wideresnet**. In both cases, 10,000 5-way 5-shot (30-query) are randomly generated. In (**a**), the variable is the LR loss, in (**b**), the DB-score, and in (**c**), the LR confidence.

**Supervised setting:** In Figure 6a, the ROC curve is built by varying a threshold value over the LR loss. When selecting a threshold value at $(0.29, 0.81)$, we obtain a confusion

matrix on the second set where 1—specificity becomes 0.14 and sensibility becomes 0.45. As both variables are lower, the chosen threshold does not apply to the second set.

**Unsupervised setting:** In Figure 6b, the ROC curve is built by varying a threshold value over the DB-score. When selecting a threshold value at $(0.30, 0.81)$, we show on the confusion matrix that 1—specificity becomes 0.64 and sensibility becomes 0.95. Here, both variables are higher. Once again, the chosen threshold does not apply to the second set.

**Semi-supervised setting:** In Figure 6c, the ROC curve is built by varying a threshold value over the LR confidence. We select the threshold value at $(0.16, 0.81)$. In the confusion matrix, 1—specificity becomes 0.18 and sensibility becomes 0.76. Both variables are similar on the two sets. So, the chosen threshold value generalizes to the second set.

The chosen threshold does not generalize in the supervised and unsupervised setting. We hypothesize that in the supervised setting, 5 shots are not enough. In the unsupervised setting, there are 30 samples per class but no label. The semi-supervised setting gets the best of both worlds.

Therefore, we perform a second experiment considering only the LR confidence in a semi-supervised setting. The advantage of the LR confidence over other measures is that it is easily interpretable. Indeed, it associates with each task the average confidence of the LR on each query sample. We propose to consider the LR confidence value as a predicted accuracy. We perform an experiment on 10,000 5-way 5-shot tasks. The average error between the real and the predicted accuracies is 2.40%. To evaluate this result, we also compute the mean absolute deviation of the real accuracies from their average. It amounts to compare the LR confidence with a naive method always predicting the same accuracy. The mean absolute deviation is 4.14%. Thus, the predictions of the LR confidence are better than the naive method.

### 5.8. Using Per-Sample Confidence to Annotate the Hardest Samples

To illustrate the usefulness of the LR confidence in practical applications, we propose a simple experiment in which we label the hardest query samples according to the LR confidence. In Figure 7, we compare what happens when labeling specific query examples, and when labeling examples at random. The data come from mini-ImageNet. Their features are extracted with **wideresnet** and diffused through a similarity graph. We observe that when the number of labeled samples is small, it is better to have a random selection. This is probably due to the fact that classes are more balanced when the annotation is randomized. Above a certain amount of labeled samples, it becomes clearly more efficient to choose the samples to label. This is not surprising as the chosen elements happen to be the ones with the lowest confidences, meaning that the remaining ones are easy to classify.

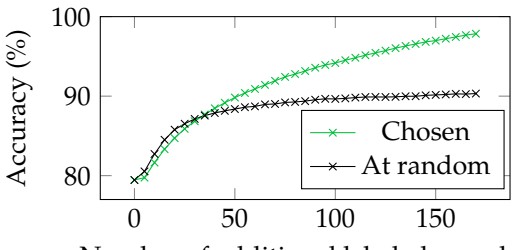

**Figure 7.** Using per-sample confidence to label data in a semi-supervised setting. We consider 5000 random 5-way 1-shot 50-query tasks. After a first training, we label either random samples or samples with a low LR confidence. In both cases, the accuracies after a second training are reported.

### 5.9. Additional Experiments

In the previous experiments, as many unlabeled samples per class were generated. In the following, we propose to explore what happens when the number of unlabeled samples is unbalanced.

In Figure 8a, we propose an experiment on 2-way 5-shot 50-query tasks. We vary the proportion of unlabeled data samples in the first class with respect to the second one. First, we observe that the correlations with the supervised measures (LR loss, similarity) decrease. Although these measures do not take into account the unlabeled samples, their accuracy is computed on an unbalanced set of samples. It might explain the decrease. Second, the correlations with the LR confidence are rather constant. As the LR confidence is directly linked to the query samples, it is more robust. Third, the correlations with the unsupervised measures (DB-score, Egv $N$) goes to 0. This is not surprising. Indeed, the DB-score measures the quality of the $N$ clusters made by a $N$-means algorithm on the unlabeled samples. As for Egv $N$, it measures to what extent a 15 nearest neighbors similarity graph computed on the unbalanced data samples is far from having $N$ connected components. When the distribution of the unlabeled samples is unbalanced, these measures on clusters/connected components no longer represent what happens in the classifier.

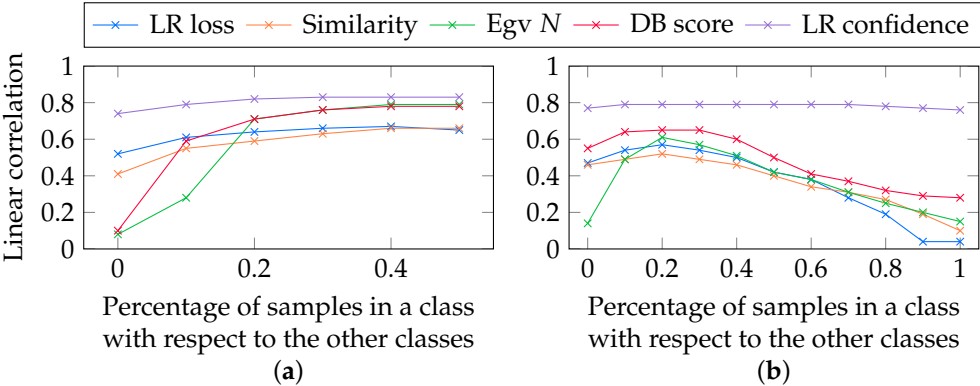

**Figure 8.** Influence of the proportion of unlabeled data samples $p$ in a class with respect to the other ones in a semi-supervised setting. The features are extracted with **wideresnet** from mini-ImageNet, and diffused through a similarity graph. We report linear correlations between the measures and the accuracy of a LR on the unlabeled samples. In (**a**), 2-way 5-shot 50-query tasks are generated. In (**b**), 5-way 5-shot 50-query. The proportion of samples in the other classes is identical. Each point is obtained over 10,000 random tasks.

In Figure 8b, the same experiment is made on 5-way 5-shot 50-query task. The proportion of unlabeled samples in a class is modified with respect to the four other classes. These four classes keep the same number of samples. When the proportion of unlabeled samples in one class is closed to 0, the problem amounts to a 4-way classification problem with balanced samples. In that case, the correlations are not really influenced. However, when the proportion goes to 1, the performance of all measures, except the LR confidence, decreases. The same reasons as in the 2-way experiment explain the results.

Finally, we propose a last experiment on the Laplacian eigenvalues measure. In Section 4, we motivated the use of the $N$-th lower eigenvalue as a measure, assuming that the $N$ classes should correspond to $N$ connected components in a graph whose edges only connect the most similar samples. However, in practice, it is expected that the number of components differ and as such more useful information could be carried by other eigenvalues. In Figure 9, we report the linear correlation in function of the number of ways where the $N$-th eigenvalue and the best one are plotted. The features of the data samples are extracted with **wideresnet** from mini-ImageNet, and diffused through a similarity graph. We observe that, in both semi-supervised and unsupervised settings, the index of the best eigenvalue is lower than $N$.

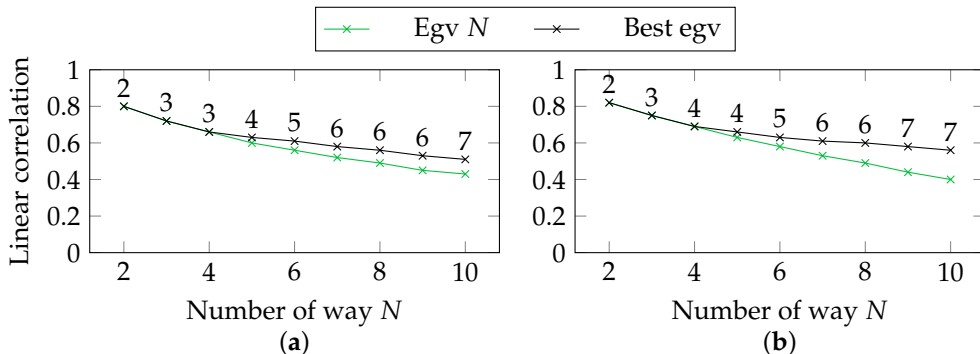

**Figure 9.** Analysis of the relevance of eigenvalues with different number of classes. In the semi-supervised setting (**a**), 5-shot 30-query tasks are generated. In the unsupervised setting (**b**), 35-query tasks. We report linear correlations between the egv $N$ and the accuracy of the LR (**a**)/the ARI of the $N$-means algorithm (**b**). In both settings, we also report the index of the eigenvalue which enables the best correlation. Each point is obtained over 10,000 random tasks.

## 6. Conclusions

In this paper, we introduced the problem of measuring the generalization performance of a few-shot classifier, taking into account the fact that we do not have a validation set. We studied several measures that we showed to be correlated to the generalization performance in various settings: supervised, unsupervised and semi-supervised. Interestingly, in the semi-supervised setting, we found a measure (LR confidence) that estimates quite well the generalization ability of a few-shot classifier despite the lack of labeled data. In the two other settings, the experiments showed that thresholds chosen within the ranges of measures to determine whether a task is easy or hard for a classifier are not applicable to new data.

We would like to investigate the relevance of other measures. Besides, in future work, we would also like to address two limitations of our method. First, the experimental results showed that the correlation between the measures and the performance of the classifier is not a robust indicator of the performance of the classifier. A better evaluation design might be defined. Second, we only performed experiments on two backbones trained on two datasets. The conclusion might be different on different backbones/datasets. Finally, we would like to inquire in more details how these findings could help in designing more efficient solutions for the few-shot problems, for example by choosing which samples to label.

**Author Contributions:** Conceptualization, formal analysis, methodology, validation, visualization, review and editing, M.B., L.B. and V.G.; investigation, software, M.B. and L.B.; data curation, writing—original draft preparation, M.B.; supervision, V.G. All authors have read and agreed to the published version of the manuscript.

**Funding:** This research received no external funding.

**Conflicts of Interest:** The authors declare no conflict of interest.

## Appendix A. Details about the Training of the Classifiers

The LR is trained on 50 epochs with the Adam optimizer [41], a learning rate of 0.01 and a weight decay of $5e - 6$. The adapted LR is trained with $\alpha = 0.75$, $k = 15$ and $\kappa = 1$, as recommended in the original paper [24]. For the $N$-means algorithm, the default implementation of scikit-learn [42] is used.

## Appendix B. Models Performance on Various Tasks

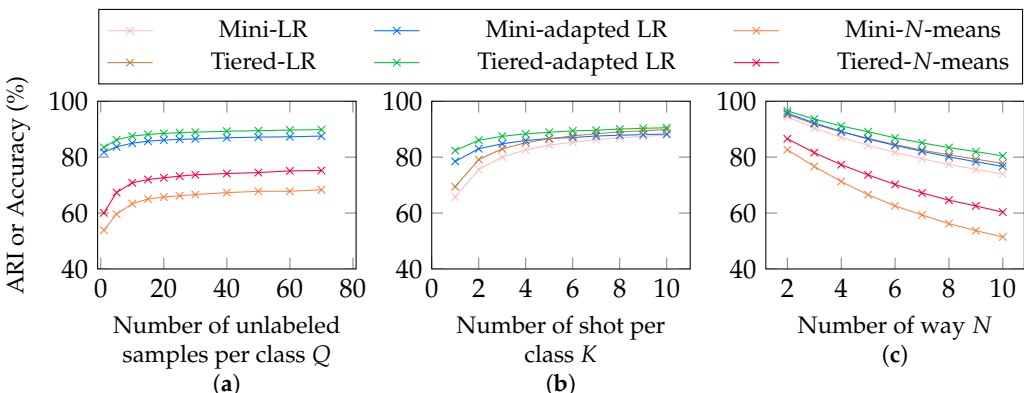

**Figure A1.** Performance of the models used in Figures 2, 4 and 5. By default, 5-way 5-shot 30-query tasks are generated. Mini/Tiered means that data come from mini-ImageNet/tiered-ImageNet. LR are the accuracies obtained in the supervised setting. Adapted LR, the accuracies obtained in the semi-supervised setting. *N*-means are the ARIs obtained in the unsupervised setting. For reasons of scale, the ARIs are multiplied by 100.

## Appendix C. Influence of the Number of Nearest Neighbors

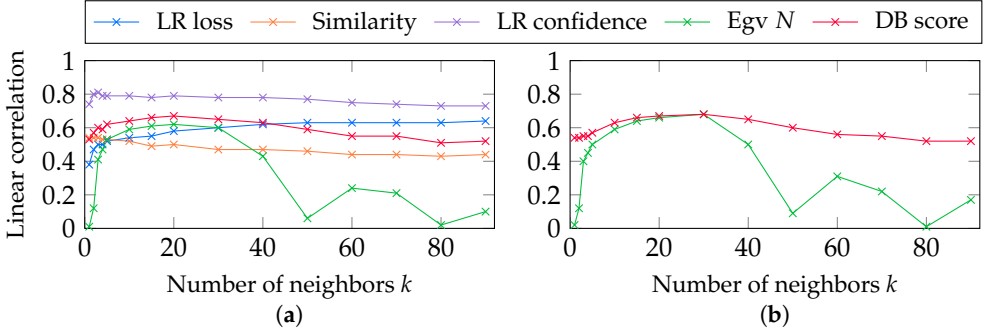

**Figure A2.** Influence of the number of neighbors *k* on the correlations. The features are extracted with **wideresnet** from mini-ImageNet and diffused through a *k*-nearest neighbors similarity graph. 5-way 5-shot 30-query tasks are generated. In (**a**), the correlations are computed between the measures and the accuracy of a LR on the unlabeled samples. In (**b**), they are computed between the measures and the ARI of a *N*-means on the unlabeled samples. Each point is obtained over 10,000 random tasks.

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
