# Peer review of "Predicting the Generalization Ability of a Few-Shot Classifier"

_information, doi:10.3390/info12010029_

Round 1

Reviewer 1 Report

  1. Authors should improve the literatures and background more effectively. 
  2. Please clarify the key contribution of the paper. 
  3. In conclusion please emphasize how the key contribution of the paper is validated
  4. In conclusion, section please explain the limitations of the method.  

Reviewer 2 Report

This work investigates the applicability of measures quantifying the generalization ability of a few-shot classifier. Three settings are considered, namely: semi-supervised, supervised and unsupervised. Various experiments are performed on vision datasets to illustrate various aspects of the problem considered.

This work addresses an interesting problem and merits publication. Although the actual technical contribution is somehow limited, it also sets the stage for future related studies. Some minor revisions should be considered:

Subsections 3.2.1-3.2.3 could be part of 3.2, without introducing separate subsections.

Subsection 4.1.2: the choice of cosine similarity instead of other similarity measures should be justified.

NQ is the number of unlabeled samples but is not defined prior to its use (it is only defined in the caption of Fig. 4).

Subsection 5.4: a comment should be included on the decrease of correlation of egvN with LR accuracy, with respect to the number of shots.

Subsection 5.7, supervised/unsupervised setting: in these two settings the authors find that the threshold derived in the first set cannot be applied in the second set. Isn’t this a major not-so-positive result that should be discussed and stressed in the conclusions?
